# Oligodendrocyte precursor cells facilitate neuronal lysosome release

Li-Pao Fang [1,2,3], Ching-Hsin Lin [3,4,14], Yasser Medlej [5,14], Renping Zhao [6], Hsin-Fang Chang [3,4], Qilin Guo [1], Zhonghao Wu [7], Yixun Su [7,8], Na Zhao [1,13], Davide Gobbo [1], Amanda Wyatt [3,9], Vanessa Wahl [3,9], Frederic Fiore [10,11], Szu-Min Tu [3,4], Ulrich Boehm [3,9], Wenhui Huang [1], Shan Bian [12], Amit Agarwal [10,11], Marcel A. Lauterbach [5], Chenju Yi [8], Jianqin Niu [7], Anja Scheller [1,3], Frank Kirchhoff [1,3] ✉ & Xianshu Bai [1,2,3] ✉

Oligodendrocyte precursor cells (OPCs) shape brain function through many non-canonical regulatory mechanisms beyond myelination. Here we show that OPCs form contacts with their processes on neuronal somata in a neuronal activity-dependent manner. These contacts facilitate exocytosis of neuronal lysosomes. A reduction in the number or branching of OPCs reduces these contacts, which is associated with lysosome accumulation and altered metabolism in neurons and more senescent neurons with age. A similar reduction in OPC branching and neuronal lysosome accumulation is seen in an early-stage mouse model of Alzheimer's disease. Our findings have implications for the prevention of age-related pathologies and the treatment of neurodegenerative diseases.

Oligodendrocyte precursor cells (OPCs) exhibit complex morphology with small cell bodies and branched, motile processes that survey their local microenvironment[1]. OPCs sense neuronal activity through the expression of glutamatergic and GABAergic receptors on their post-synaptic terminals[2–5]. Under pathological conditions, including acute brain injury or epilepsy, where neuronal activities are often increased[6,7], OPCs become hypertrophic, elevating their morphological complexity[8,9]. Besides potential differentiation into oligodendrocytes, whether OPCs exert other functions through increasing morphological complexity remains unknown.

Recent studies unveil diverse pathways for OPC-neuron interactions, particularly between OPC processes and different compartment of neurons[10], including GABA release to neurons in the hippocampus via synaptic complexes[11], synaptic pruning by phagocytosing axonal terminals during development[12], and putative modulation of action potential propagation through contacts at the nodes of Ranvier[13]. However, considering the complex morphology of OPCs and the densely populated neurons in the brain, we asked whether there is a mode of communication between OPCs and neurons potentially through alternative physical contacts such as direct interactions with neuronal somata.

[1]Molecular Physiology, Center for Integrative Physiology and Molecular Medicine (CIPMM), University of Saarland, 66421 Homburg, Germany. [2]State Key Laboratory of Natural Medicines, Department of Pharmacology, School of Pharmacy, China Pharmaceutical University, 211198 Nanjing, China. [3]Center for Gender-specific Biology and Medicine (CGBM), University of Saarland, 66421 Homburg, Germany. [4]Cellular Neurophysiology, CIPMM, University of Saarland, 66421 Homburg, Germany. [5]Molecular Imaging, CIPMM, University of Saarland, 66421 Homburg, Germany. [6]Biophysics, CIPMM, University of Saarland, 66421 Homburg, Germany. [7]Research Centre, Seventh Affiliated Hospital of Sun Yat-sen University, 518107 Shenzhen, China. [8]Department of Histology and Embryology, Chongqing Key Laboratory of Neurobiology, Brain and Intelligence Research Key Laboratory of Chongqing Education Commission, Third Military Medical University, 400038 Chongqing, China. [9]Experimental Pharmacology, Center for Molecular Signaling (PZMS), Saarland University School of Medicine, Homburg, Germany. [10]The Chica and Heinz Schaller Research Group, Institute for Anatomy and Cell Biology, Heidelberg University, Heidelberg, Germany. [11]Interdisciplinary Center for Neurosciences, Heidelberg University, Heidelberg, Germany. [12]Institute for Regenerative Medicine, State Key Laboratory of Cardiology and Medical Innovation Center, Shanghai East Hospital, Frontier Science Center for Stem Cell Research, School of Life Sciences and Technology, Tongji University, Shanghai, China. [13]Present address: Institute of Anatomy and Cell Biology, University of Saarland, 66421 Homburg, Germany. [14]These authors contributed equally: Ching-Hsin Lin, Yasser Medlej. ✉e-mail: frank.kirchhoff@uks.eu; xianshu.bai@uks.eu

Here, we show that OPCs form direct contact on neuronal somata and the prevalence of such contact is neuronal activity-dependent. Further ultrastructural and functional analysis of these contacts reveals that OPC-neuron contact facilitates the exocytosis of neuronal lysosomes, thereby influencing neuronal metabolism and senescence. Our findings portend an OPC-based clinical strategy for preventing aging-related pathologies and the treatment of neurodegenerative diseases in which neuronal lysosomal function is compromised.

## Results

### OPCs prefer active neurons for process-somata contact

To explore the spatial relationship between OPC processes and neurons, we performed triple immunostaining for PDGFRα, GFP (recognizing EYFP) and NeuN in adult NG2-EYFP (NG2[EYFP]) mouse brains (Fig. 1A). The GFP⁺PDGFRα⁺ OPCs exhibited a ramified morphology, with several processes making contact with NeuN⁺ neuronal somata (Fig. 1B, C). These contacts were commonly observed in the majority, if not all, of the neurons (90.9–99.3%) across various regions of gray

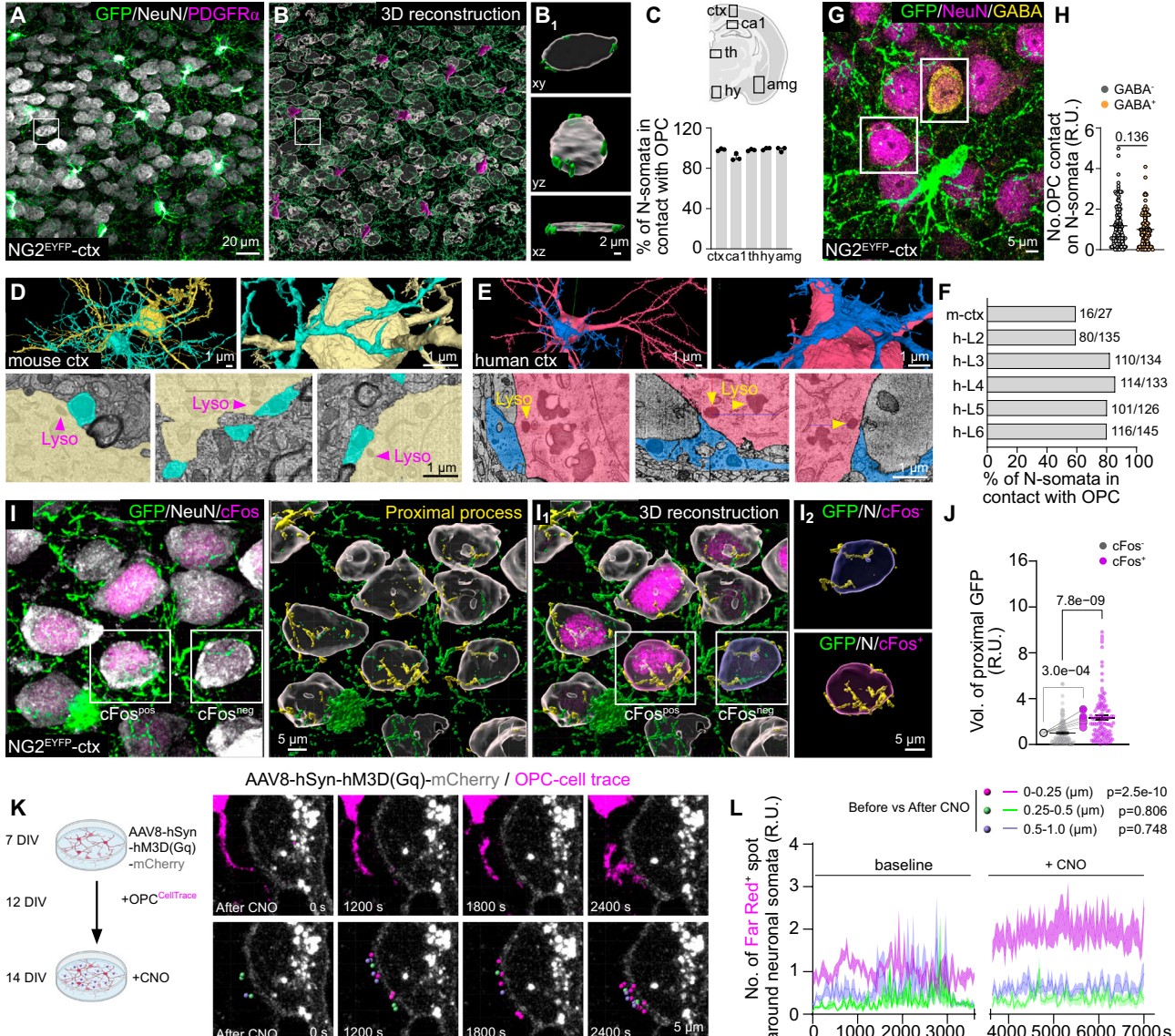

**Fig. 1 | OPC processes contact neuronal somata with a preference for active neurons. A, B** Immunostaining and 3D reconstruction of OPCs and neurons with GFP, PDGFRα and NeuN antibodies in coronal brain slices of NG2[EYFP] mice. **B₁** Exemplary images of process-somata contact (PSC) between OPCs and neurons from the boxed area of **B**, shown in three directions. **C** Percentage of neurons with PSCs in the cortex (ctx), hippocampal CA1 region (ca1), thalamus (th), hypothalamus (hy) and amygdala (amg) (*N* = 3 mice). **D–F** Ultrastructural images (**D, E**) and quantification (**F**) of PSCs in mouse (**D**) and human cortex (**E**). **G, H** Immunostaining and quantification of OPC (GFP⁺) contacts on GABAergic (GABA⁺) and non-GABAergic (GABA⁻) neurons in NG2[EYFP] mice (127 GABA⁻ neurons and 109 GABA⁺ neurons from 5 mice, normalized to GABA⁻ neurons, two-sided unpaired *t* test). **I** Immunostaining of OPCs and neurons with GFP and NeuN in the cortex of NG2[EYFP] mice. Neuronal activity was indicated with cFos (magenta) immunoreactivity and contacting processes were indicated in yellow. **I₁** 3D-reconstruction of neuronal somata and OPCs. OPC processes contacting neuronal somata were indicated in yellow. **I₂** Exemplary cFos⁺ and cFos⁻ neurons with OPC contacts. **J** Quantification of relative volume of OPC processes (GFP⁺) contacting cFos⁺ and cFos⁻ neurons (normalized to cFos⁻ cells). (117 cFos⁻ cells and 118 cFos⁺ cells from 8 mice, two-sided unpaired t-test for cells (small dots) and two-sided paired t-test for mice (large dots)). **K** Live-imaging of neuron (hM3D(Gq)-mCherry⁺)-OPC (CellTrace⁺) interaction before and after CNO application. **L** The relative number of processes contacting (0–0.25 μm), near (0.25–0.5 μm) and far from (0.5–1.0 μm) the neurons were quantified and compared before and after CNO application (5 cells/5 independent experiments, one-way ANOVA, Multiple comparison). R.U.=relative units, normalized to the controls (5 cells/5 independent experiments, in each experiment OPCs are prepared from 4 pups). Data are shown as mean ± SEM in **C, H, J,** and **L**. Source data are provided as a Source Data file. Created in BioRender. Fang, L. (2025) https://BioRender.com/n88b804.

matter, including the cortex, hippocampus, thalamus, hypothalamus, and amygdala, in both developing and adult brains (Fig. 1C, Supplementary Figs. 1, 2). The prevalent presence of OPC process-somata contacts (PSCs) was further confirmed through ultrastructural analysis of mouse visual cortex[14] (at least 59 %) and human cortex[15] (at least 59–86% in cortical layer 2-6), using publicly available databases (Fig. 1D–F).

During the analysis, we noticed a variability in the number of contacts among neurons. However, both excitatory (NeuN+GABA-) and inhibitory neurons (NeuN+GABA+) had a similar contact frequency (Fig. 1G, H). Given that OPCs sense neuronal activity[10,16], we next explored the correlation between neuronal activity and the number of PSCs. We distinguished between active and less/non-active neurons based on their immunoreactivity to cFos (Fig. 1I), an immediate-early gene and well-established marker of cellular activity[17,18], and labeled OPC processes with GFP antibodies in NG2^EYFP mice or NG2 antibodies in the wild type mice (Supplementary Fig. 3). We found that cFos+ active neurons had approximately twice as many PSCs compared to cFos- less/non-active neurons (Fig. 1I, J; Supplementary Fig. 3), suggesting that OPCs preferentially contact active neurons. To further investigate the causal link between neuronal activity and PSC formation, we employed a chemogenetic approach to activate cortical neurons (Supplementary Fig. 4) or OPCs (Supplementary Fig. 5). In the ipsilateral cortex of mice with increased neuronal activity, OPCs increased morphological complexity and neurons formed three times more PSCs than those in the contralateral side (Supplementary Fig. 4). However stimulation of OPCs did not change the frequency of PSC formation on neurons (Supplementary Fig. 5), suggesting that PSC formation is primarily driven by neuronal activity. To substantiate this observation, we used neuron-OPC co-culture system, where neurons were activated with AAV8-hSyn-hM3D(Gq)-mCherry and CNO (Clozapine N-oxide) (Fig. 1K). In response to the neuronal activation, we followed the recruitment of OPC processes towards neuronal somata by confocal laser-scanning microscopy (Supplementary Videos 1, 2). Based on the distance between OPC processes and the neuronal surface, processes were classified into three subgroups (Fig. 1L): 0–0.25 μm (proximal, magenta), 0.25–0.5 μm (medial, green), and 0.5–1.0 μm (distal, blue), with the relative volume of processes in each category quantified. While neuronal stimulation (Supplementary Fig. 6) increased the volume of proximal processes (magenta, Fig. 1K, L), that of medial and distal processes remained unchanged. These results strongly suggest that neurons attract OPC processes to their somata in an activity-dependent manner.

## OPC contact facilitates neuronal lysosome release
To explore the physiological significance of PSCs, we further analyzed the ultrastructural images and observed electron-dense organelles in neurons, highly likely lysosomes, closely positioned near the contact sites (distance ranging 60–90 nm and 100–400 nm in mouse and human cortex, respectively) (Fig. 1D, E). The identity of these organelles was confirmed as lysosomes (LAMP1+), visualized with stimulated emission depletion (STED) super-resolution microscopy, positioned near the contact site (Fig. 2A, B). Lysosomes vary in size, with smaller, ready-to-be-released lysosomes typically located closer to the plasma membrane[19]. Thus, we quantified the volumes of individual neuronal lysosomes and measured their shortest distance to the contact site (Fig. 2C, D). A strong correlation was found between lysosome size and proximity to the opposing OPC surface, with smaller lysosomes positioned closer to the contact site (Fig. 2D, E), suggesting that PSCs may promote lysosome exocytosis. To address this possibility, we imaged neuronal lysosome trafficking in a neuron-OPC co-culture (Fig. 2F). Lysotracker, a dye that labels low-pH organelles such as lysosomes[20], was applied before imaging. We observed lysosomes being recruited to the PSC site (Fig. 2F), accompanied by a decrease in lysosomal fluorescence intensity (FI). A few minutes later, this

lysosome moved back into the cytosol with its FI restored, presumably being recycled (Fig. 2F) (Supplementary Video 3), suggesting this lysosome may have experienced exocytosis at the OPC-contact site. To confirm and quantify the exocytosis, we added Alexa647-anti-LAMP1 antibodies to the medium, which bind to the luminal domain of Lamp1, exposed during lysosomal exocytosis. Given that active neurons form more PSCs, we increased neuronal activity with AAV8-hSyn-hM3D(Gq)-mCherry combined with CNO (Fig. 2G) (Supplementary Videos 4–6). Neurons with OPC contacts (N + OPC) exhibited significantly higher levels of lysosome exocytosis compared to neurons co-cultured with OPCs without contacts (N + OPC/without contact) or neurons cultured alone (Fig. 2G, Supplementary Fig. 7). Notably, exocytosis was more frequent near OPC contact sites (<1 μm), suggesting OPC contacts facilitate neuronal lysosome exocytosis.

To investigate the impact of PSC-mediated neuronal lysosome exocytosis, we employed a mouse model with OPC-specific deletion of L-type voltage-gated calcium channels Cav1.2 and Cav1.3[21] (Supplementary Fig. 8). In these OPC-Cav1.2/Cav1.3 double knockout (dKO) mice, OPCs exhibit simpler morphology and form less PSCs with neurons (Fig. 2I; Supplementary Fig. 8). First, we assessed lysosome exocytosis using acute brain slices of these mice which were collected at four weeks after AAV-PHP.eB-hSyn-Lamp1-mScarlet (labeling neuronal lysosomes) injection and incubated with Alexa647-anti-Lamp1 antibodies for 30 min. The slices were imaged with confocal laser scanning microscopy and the number of mScarlet+Lamp1+ lysosomes within the neuronal somata were quantified. As expected, the dKO mice showed decreased number of lysosomal exocytosis in neurons compared to the control mice (Fig. 2H), confirming that reduced contacts lead to decreased exocytosis. Concomitantly, we observed that the number and the size of the lysosomes in neurons of dKO mice were increased compared to control mice (Fig. 2I_2, I_3). Also, the lysosomal distribution in neuronal somata was scattered with an increased distance to the surface of OPCs, in contrast to the control condition, where neuronal lysosomes were enriched at the contact site (Supplementary Fig. 8J, K). Using an acute OPC depletion mouse model (PDGFRα-CreERT2 x Rosa26-flSTOPfl-DTA[22]), we observed more and larger lysosomes in the neurons of DTA mice compared to that of age-matched control animals (Fig. 2J).

Taken together, these results suggest that PSCs affect neuronal lysosome exocytosis. Fewer OPC contacts on neuronal somata led to impaired lysosome exocytosis and aberrant accumulation of lysosomes.

## Impaired process-soma contact links to senescence and degeneration
Lysosomes, as degradative organelles, play a role in digesting diverse biomolecules, including lipid droplets (LDs). To investigate whether the abnormal accumulation of lysosomes corresponds to lysosomal dysfunction, we examined LD density and volume in cortical neurons of both control and dKO mice by employing LD staining combined with NeuN (Fig. 3A). The LD volume in neurons of dKO mice was about three times larger than that in the control group, indicating an accumulation of LDs and impaired lysosomal function in dKO mouse neurons. Given that excessive LD accumulation can impact cellular function, we proceeded to analyse neuronal activity using cFos immunostaining. In the mutant mouse brain, including cortex, hippocampus, and hypothalamus, the total number of cFos+ cells was higher compared to the control group (Fig. 3B), indicating higher activation of neurons in the mutant mouse brain.

Transcriptomic analysis suggested that the expression of genes involved in the arrangement of cytoskeleton was altered in the OPCs of dKO mice, in line with our observation of reduced morphological complexity of OPCs, while neurons in the mutant mice exhibited a change in the expression of genes involved in the pathways of exocytosis, lysosome function and lipid metabolism, etc., again

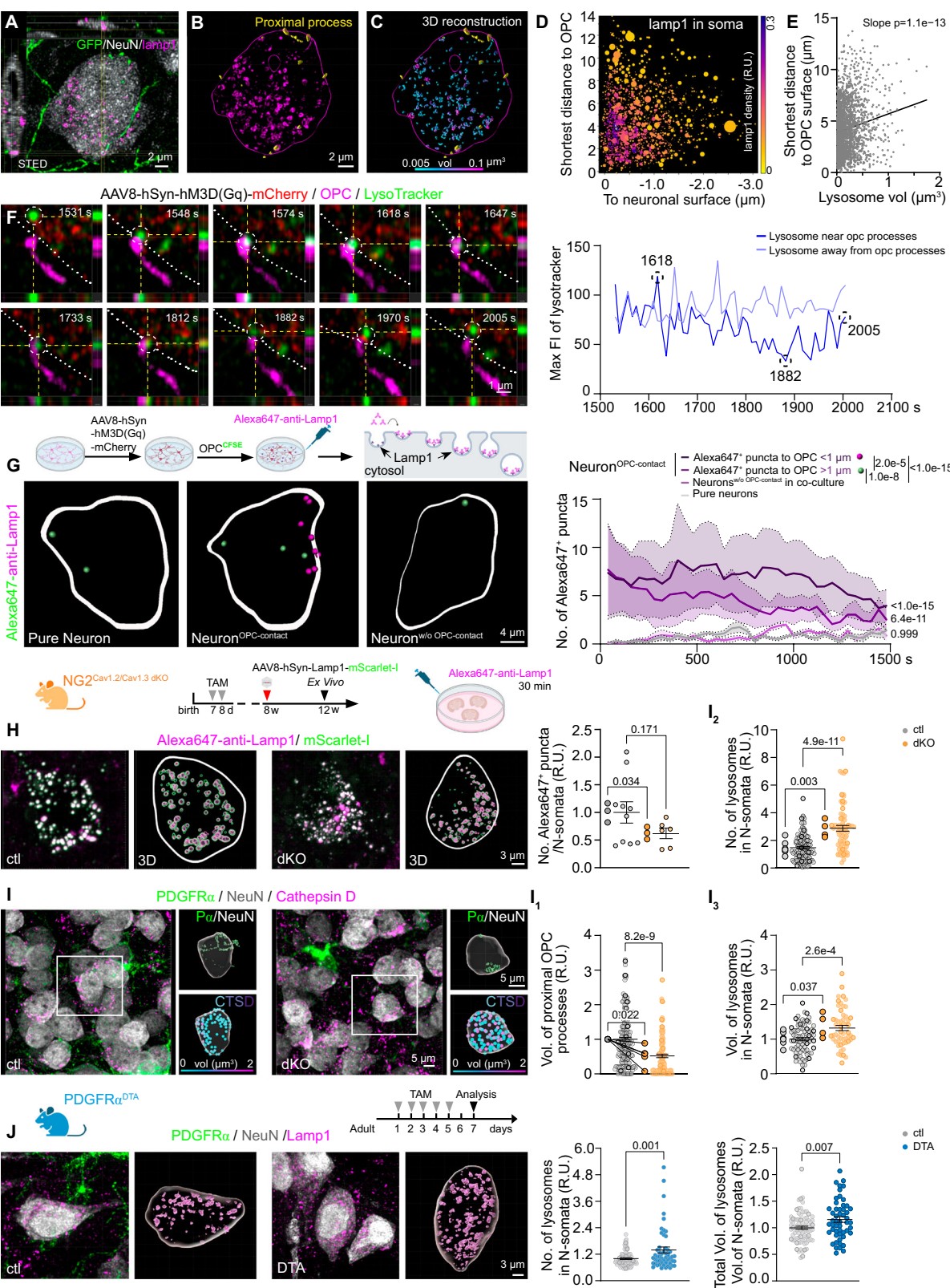

highlighting neuron-OPC contact is linked to neuronal metabolism. In addition, we also observed that in both OPCs and neurons of mutant mice, the genes involved in neurodegeneration and senescence were upregulated, particularly those involved in Alzheimer's disease (AD), eg. *Apoc1, Pla2g4a, and Cd74* (Fig. 3C–E, Supplementary Figs. 9, 10). As the dKO mice do not develop AD, we first checked whether neuronal senescence would be affected by the impaired OPC-neuron contact.

We performed immunostaining of p16[INK4A], an established senescence marker[23] in the neurons of control and dKO mice (Fig. 3F), and found about 20-fold increase of neuronal senescence in the 12-month-old mutant mouse brain (Fig. 3G), compared to age-matched control animals. These data suggest that neuron-OPC contacts are crucial components for neuronal metabolism and may prevent neuronal senescence. Next, we investigated whether OPC-neuron contact is

**Fig. 2 | OPC contact facilitates neuronal lysosome release at the contact site.** **A**–**C** Immunostaining and 3D-reconstruction of neurons, OPCs and lysosomes in NG2[EYFP] mouse cortex using NeuN, GFP and Lamp1 antibodies. **B** OPC processes contacting neuronal somata were indicated in yellow. **C** Neuronal lysosomes shown in colors based on their volume. **D** Display of lysosome volume (indicated by circle size) and their proximity to OPC and neuronal surfaces (5925 lysosomes from 17 cells/3 mice). The color indicates the frequency of lysosomal appearance at corresponding location. **E** Correlation of lysosome volume and proximity to OPC surface (2077 lysosomes from 17 cells/3 mice). **F** Live-imaging of a lysosome (dotted circle) trafficking in neuron-OPC co-culture and analysis of maximum fluorescence intensity (MAX FI) of lysosomes over the period. Dotted line indicates neuronal membrane. **G** Live-imaging and quantification of lysosome exocytosis in neurons from pure neuronal culture and neurons with (neuron[OPC-contact]) or without OPC contact (neurons[w/o-OPC-contact]) from co-culture. In neurons[OPC-contact], the Alexa647[+] puncta were classified into proximal (<1 μm, magenta) and distal (>1 μm, green) based on their distance to OPC contact site (Pure neuron=9 cells/4 independent experiments; neurons[OPC-contact] = 9 cells/4 independent experiments,

neurons[w/o-OPC-contact] = 3 cells/3 independent experiments; 4 pups/experiment, two-way ANOVA, multiple comparisons). **H** Lysosome exocytosis in acute brain slices was imaged and the number of Alexa647[+] puncta in each neuron was quantified (ctl = 10 cells/3 mice; dKO=6 cells/3 mice). **I** Immunostaining and 3D-reconstruction of OPCs (PDGFRα[+]), neurons (NeuN[+]) and lysosomes (Cathepsin D, CTSD[+]). **I₁** Quantification of relative volume of OPC processes contacting neuronal somata (ctl = 190 cells/5 mice, dKO = 116 cells/5 mice). **I₂, I₃** Analysis of the relative number and volume of neuronal lysosomes. (ctl = 112 cells/7 mice, dKO = 74 cells/4 mice). **J** Immunolabeling and 3D reconstruction of neurons, OPCs and lysosomes with NeuN, PDGFRα and Lamp1 antibodies. The relative number and volume of neuronal lysosomes were quantified. (ctl = 80 cells/5 mice, DTA = 52 cells/4 mice). In (**H**–**J**), small dots represent cells while big dots represent mice, two-sided unpaired t-tests were used. R.U. = relative units, normalized to the controls. Data are shown as mean ± SEM in (**G**–**J**). Source data are provided as a Source Data file. Created in BioRender. Fang, L. (2025) https://BioRender.com/a0Op319 and https://BioRender.com/v85j019.

relevant to AD pathology. We observed that OPCs immunolabeled with PDGFRα exhibited a simpler morphology (Fig. 3H–J) and less contacts to neuronal somata (Fig. 3K, L) in the early-stage Tg2576 AD mouse model[24]. Further analysis of neuronal lysosomes and their distance to the contact sites using PDGFRα, NeuN and Cathepsin D triple immunostaining (Fig. 3K) showed increased numbers of lysosomes in neurons (Fig. 3L) and the larger distances between lysosomes and the OPC surface in the cortex of Tg2576 mice (Fig. 3M, N). These results suggest that PSC-mediated lysosome release may play a protective role against neuronal senescence and degeneration.

In conclusion, our study highlights the crucial role of OPCs in modulating neuronal lysosome release through direct contact with neuronal somata. The disruption of the process-somata contacts between OPCs and neurons leads to aberrant neuronal metabolism, neuronal senescence, and may ultimately neurodegeneration. These findings underscore the significance of OPC-neuron interactions in maintaining proper neuronal function and provide insights into potential mechanisms underlying neurodegenerative conditions.

## Discussion

Emerging evidence supports the notion that OPCs play a key role in modulating neuronal function through various mechanisms[3,10,12,14,25]. Our study highlights the critical role of OPCs in regulating neuronal lysosome exocytosis via direct contact with neuronal somata. Lysosomal function is closely linked to metabolic[26] and numerous neurological diseases[27]. In neurodegenerative conditions, such as Alzheimer's disease, lysosomal dysfunction in neurons is observed even before the onset of neurological symptoms[28]. Hence, early intervention targeting lysosomal dysfunction, particularly through OPC-neuron interactions, may help prevent neuronal impairment and neurodegeneration.

The process-somata contacts between OPCs and neurons may also serve as avenues for signal exchange. Beyond the lysosome's canonical role in cellular 'waste' disposal, more and more studies highlight the role of lysosomes as signaling hubs[29]. Lysosomes contain high levels of ATP, which can be released to the cellular periphery[30]. Coincidently, OPCs express a series of purinergic receptors[31], including P2Y1[32], P2X7[33], P2Y12 receptors, etc. For instance, activation of P2X7 and P2Y1 receptors in OPCs by ATP has been implicated in promoting OPC migration in vitro[33,34]. This raised the possibility that ATP derived from neuronal lysosomes activates P2Y1 and P2X7 receptors, thereby recruiting OPC processes to neuronal somata. However, the temporal sequence of events, whether the process-soma contact precedes lysosome fusion or vice versa, remains unclear. Of note, in the Cav1.2/Cav1.3 double knockout (dKO) mutant mice where OPC-neuron contact is reduced, or in the mice where OPCs were acutely depleted, aberrant lysosome accumulation in neurons is observed, with

lysosomes positioned farther from OPC contact sites. Hence, OPC contact emerges as a critical factor for lysosomal release from neurons.

In conclusion, our study demonstrates that OPCs establish proximal junctions with neuronal somata, which are important for modulating the exocytosis of neuronal lysosomes and neuronal metabolism.

## Materials and methods
### Ethics statement

The majority of the experiments were carried out at the University of Saarland in strict accordance with recommendations of European and German guidelines for the welfare of experimental animals. Animal experiments were approved by Saarland state's "Landesamt für Gesundheit und Verbraucherschutz" in Saarbrücken/Germany (animal license numbers: 65/2013, 12/2014, 34/2016, 36/2016, 03/2021, 07/2021, 08/2021, Perfusion-2023, VM2024-03 and 17/2023) and 'Laboratory animal welfare and ethics committee' of the Third Military Medical University, China (SYXK (YU) 2022-0018).

### Animals

Majority of the mouse lines were maintained in C57BL/6N background and most of them were housed at the animal facility of the CIPMM. Mice were kept on a 12 h (h) light/dark cycle at 20-24°C with 55-65% humidity and fed a breeding diet (V1125, Sniff) *ad libitum*. To better observe OPC morphology, NG2-EYFP knock-in mice (NG2[EYFP])[35] were used. To conditionally knock out (KO) Cav1.2 and Cav1.3 in oligodendrocyte precursor cells (OPCs), NG2-CreER[T2] mice were crossbred to Cav1.2[fl/fl] (flanking exon 14 and 15 of *cacna1c*) and Cav1.3[fl/fl] mice (flanking exon 2 of *cacna1d*) (NG2[Cav1.2/Cav1.3 dKO])[21]. PDGFRα-CreER[T2] x Rosa26-STOP-[fl]DTA[fl] mice[22,36] were maintained in the animal facility of Chongqing Key Laboratory of Neurobiology, Third Military Medical University, Chongqing, China. Brain slices of the Tg2576 Alzheimer's disease mouse model were kindly provided by Dr. Manuel Buttini (Neuropathology, Luxembourg Centre for Systems Biomedicine, University of Luxembourg, Luxembourg).

### Tamoxifen administration

Tamoxifen (Carbolution, Neunkirchen, Germany) was dissolved in Miglyol®812 (Caesar & Lorentz GmbH, Hilden, Germany) to a final concentration of 10 mg/ml and administered intraperitoneally (100 mg/kg body weight) for two consecutive days at postnatal day (P) 7 and 8 or for five consecutive days at the age of 4 weeks (Fig. 3F)[3,37]. For PDGFRα-CreER[T2] x Rosa26-STOP-[fl]DTA[fl] mice, 100 μl of tamoxifen (30 mg/ml, MCE, HY-13757A) of tamoxifen was administered via gastric lavage for five consecutive days and analyzed two days after the last tamoxifen application.

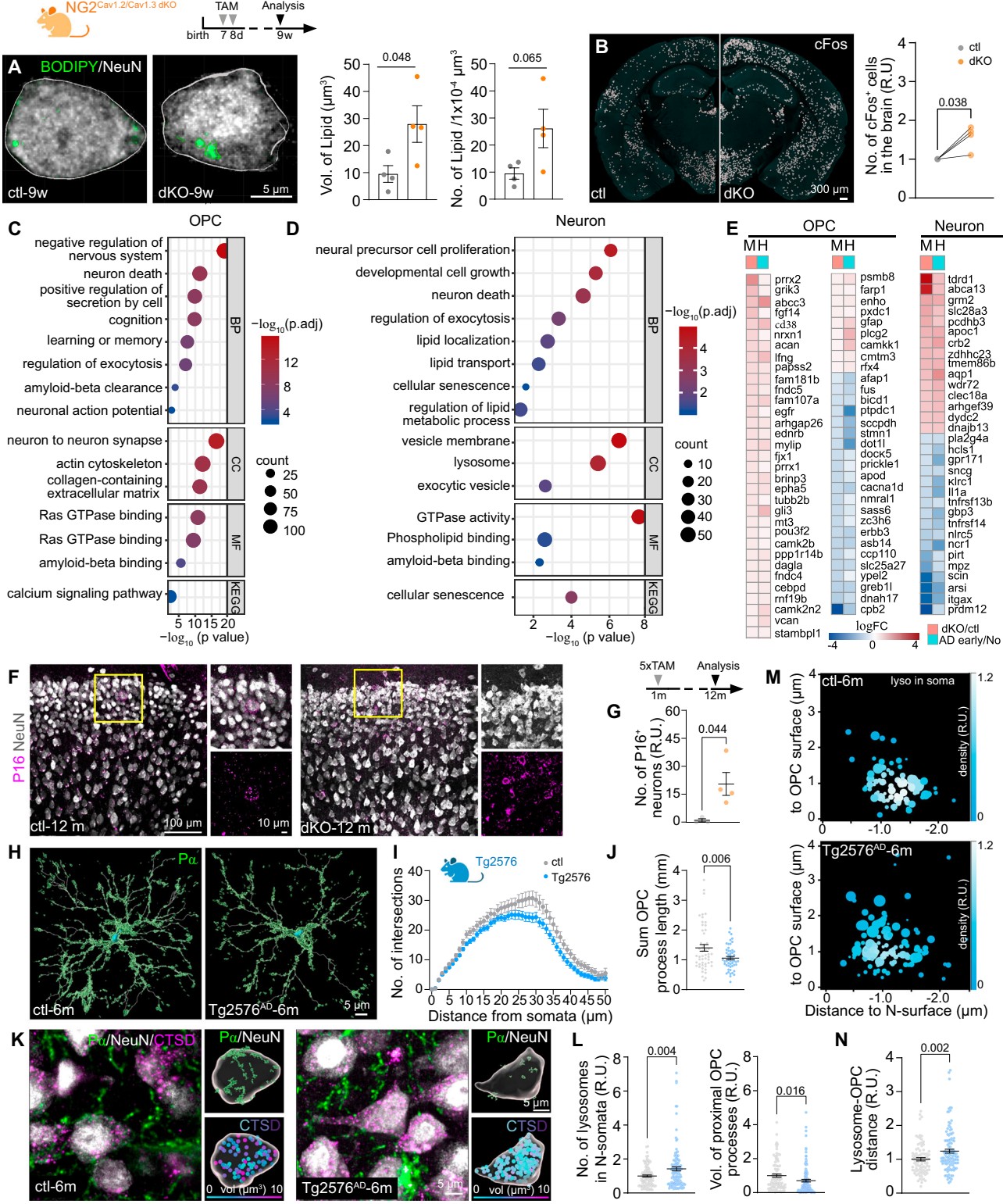

## Magnetic activated cell sorting of OPCs and neurons

Mice were initially anaesthetized with ketamin/Xylazin (100 and 12.5 mg/kg bodyweight, respectively; diluted in saline, i.p.) and double anesthetic dose (Ketamin/xylazine 200 and 25 mg/kg bodyweight) was given after the loss of consciousness. Mice were perfused with cold Hank's balanced salt solution lacking $Ca^{2+}$ and $Mg^{2+}$ (HBSS, H6648, Gibco), and cortices were dissected in ice-cold HBSS. After removing debris (130-107-677, Miltenyi Biotec), cells were resuspended in 1 mL of "re-expression medium" containing NeuroBrew-21 (diluted 1:50 in

MACS Neuro Medium) (130-093-566 and 130-093-570, Miltenyi Biotec) and 200 mM L-glutamine (diluted 1:100, G7513, Sigma) at 37 °C for 30 min. Subsequently, cells were incubated with Fc-receptor blocker for 10 min at 4 °C (provided with the CD140 MicroBeads kit), followed by a 15 min incubation with a 10 µl microbeads mixture containing antibodies against CD140 (130-101-502), NG2 (130-097-170), and O4 (130-096-670) in a 1:1:1 ratio at 4 °C. After passing through the magnetic column, the cells attached to the column was washed out and collected.

**Fig. 3 | Reduced OPC-neuron contacts induce an accumulation of lysosomes and functional impairment in neurons. A** Immunostaining and quantification of lipid droplets in cortical neurons of control (ctl) and double knockout (dKO) mice with NeuN and BODIPY antibodies, respectively (4 mice per group). **B** Immunostaining and quantification of cFos$^+$ cells in the whole brain of ctl and dKO mice (4 mice per group). **C, D** Dot plot of enriched gene ontology terms for biological processes (BP), cellular components (CC), molecular functions (MF), and KEGG pathways in genes from OPCs and neurons of dKO mice. **E** List of differentially expressed genes in OPCs and neurons of the dKO mice (M, bulkRNA-seq) and AD patients in early pathology (H, scRNA-seq). **F, G** Immunostaining and quantification of senescent neurons with NeuN and p16$^{INK4A}$ (P16) at 12 months of age (ctl = 3 mice, dKO = 4 mice). **H** 3D reconstruction of OPCs immunolabelled with PDGFRα (Pα) at 6 months of age. **I, J** Morphological analysis of OPCs for the number of process intersections (**I**) and total process length (**J**) (ctl = 51 cells/3 mice, Tg2576 = 52 cells/3 mice). **K** Immunostaining and 3D reconstruction of OPCs (Pα), neurons (NeuN) and lysosomes (Cathepsin D, CSTD). **L** Quantification of the relative volume of OPC processes in contact with neuronal soma (ctl = 91 cells/3 mice, Tg2576 = 117 cells/3 mice) and the number of neuronal lysosomes (ctl=87 neurons/5 mice, Tg2576 = 115 neurons/5 mice). **M, N** Quantification of the relative mean distance between neuronal lysosomes and OPC surface (ctl = 91 cells/3 mice, Tg2576 = 117 cells/3 mice). **M** The size of the lysosomes is represented by the size of the filled circles, while the color indicates the frequency of each size in the total analysis. R.U. = relative units, normalized to the controls. Data are shown as mean ± SEM in (**A, G, I, J, L,** and **N**). Two-sided unpaired $t$ test was used in (**A, B, G, I, J, L,** and **N**). Source data are provided as a Source Data file. Created in BioRender. Fang, L. (2025) https://BioRender.com/f32o880.

Neuronal isolation was performed with negative sorting following the manufacturer's instruction (130-126-603, Miltenyi Biotec). Single-cell suspension is prepared from the cortex, incubated with mixture of ACSA-2, CD31, CD11b and O4 beads. The mixture was through the magnetic column and the flow was collected.

## Primary culture of cortical neurons and OPCs

**Primary cortical neuron culture.** One C57BL/6N mouse aged P0–1 was decapitalized, cerebral cortices were carefully dissected, and the meninges were removed in ice-cold Earle's Balanced Salt Solution (EBSS, Gibco). Subsequently, the tissue was finely minced and subjected to digestion with 35 units/ml papain (Worthington, NJ) for 45 min at 37 °C, followed by gentle mechanical trituration. The resulting cell suspension was filtered through a 70 μm cell strainer (Greiner Bio-One). Dissociated cell suspensions were then carefully seeded on 25-mm glass coverslips in 6-well plates ($3 \times 10^5$ cells/well) or 12-mm glass coverslips in 24-well plates ($1 \times 10^5$ cells/well). Prior to seeding, the glass coverslips were pre-coated with a solution comprising 17 mM acetic acid, poly-D-lysine (Sigma, St. Louis, MI, USA, P6407), and collagen I (Gibco, A1048301). Neurons were cultured in Neuronal-A (NBA) medium supplemented with 10% FCS, 1% Penicillin/Streptomycin, 1% GlutaMAX, and 2% B-27 supplement (Gibco). On the second day, the medium was replaced with fresh medium to eliminate residual cell debris from the initial cell preparation. Unless specified otherwise, neurons were maintained in the medium for 8–14 days at 37 °C with 5% CO$_2$ before experimental procedures.

**AAV expression in primary cortical neurons.** At 7 DIV, AAV8-hSynapsin1-hM3D(Gq)-mCherry ($\geq 2 \times 10^{11}$ vg/ml) was added to the 6-well plate (2 ml per well). To activate primary cortical neurons, 40 μM of CNO was added after baseline recording.

**OPC sorting for cell culture.** OPCs were obtained from neonatal C57BL/6N or NG2$^{EYFP}$ mouse brains (age <P7) using a modified MACs procedure. Briefly, four mice were decapitalized and cortices were dissected in cold HBSS (without Ca$^{2+}$/Mg$^{2+}$), followed by dissociation using the Neural Tissue Dissociation Kit (130-092-628, Miltenyi Biotec). The dissociation process was stopped by adding 10 ml of DMEM high glucose medium (11965092, Fisher Scientific) with 1% horse serum (HS, Fisher Scientific). After a brief centrifuge at 400 g, cells were resuspended in DMEM and filtered through 70 μm and 40 μm cell strainers successively. The collected cells were then resuspended in basic medium (NeuroBrew-21 in MACs Neuro Medium (1:50) (130-093-566 and 130-093-570, respectively, Miltenyi Biotec) with 200 mM L-glutamine (1:100, G7513, Sigma)). The cells were incubated at 37 °C for 1.5-3 hs, followed by centrifugation at 4 °C and resuspension with 134 μl/brain of DMEM + 1% HS, along with a mixture of microbeads (20 μl/brain) conjugated with antibodies against CD140 (130-101-502), NG2 (130-097-170), and O4 (130-096-670) in a 1:1:1 ratio for 15 min at 4 °C. The cells were then resuspended in 2 ml DMEM + 1% HS/brain, followed by centrifugation at 400 g for 10 min at 4 °C. In subsequent steps, DMEM + 1% HS was used instead of MACs buffer, following the manufacturer's instructions (130-101-502). Finally, 5 ml of proliferation medium (containing 4.2 μg/ml Forskolin (F6886, Sigma), 10 ng/ml CNTF (450-13, Pepro Tech), 10 ng/ml PDGF-AA (100-13A, Pepro Tech), 1 ng/ml NT-3 (450-03, Pepro Tech) in basic medium) was added to flush out OPCs. OPCs isolated from C57Bl/6 N were further labeled with CellTracer CFSE (C34570, Invitrogen) or CellTracer FarRed (C34554, Invitrogen). Cells were seeded in plates pre-coated with Poly-L-Lysine (Merck) at a density of $1 \times 10^5$ cells/well (24-well plate) or $3 \times 10^5$ cells/well (6-well plate).

**OPC-neuron co-culture.** To investigate lysosome communication between OPCs and neurons, at 10-12 DIV of neuronal culture, OPCs were seeded at a density of $5 \times 10^4$ cells/well of 24-well plate or $1.5 \times 10^5$ cells/well in 6-well plate on top of the neurons. The co-cultures were maintained in optimized medium (1:1 of OPC proliferation medium and neuron NBA culture medium) for additional 2–3 days throughout the experiments.

## Immunohistochemistry

Mice were anaesthetized as described above and perfused with PBS followed by 4 % PFA and post-fixed at 4 °C overnight. For NG2 immunostaining, we perfused animals with 2% PFA and post-fixed for 4 h at 4°C. After post-fixation, coronal slices in 40 μm thickness were prepared using a Leica VT1000S. Free floating slices were incubated blocking buffer (5% HS with 0.5% Triton in PBS) for 1 h at R.T. followed by primary antibody incubation. The brains of PDGFRα-CreERT2 x Rosa26-STOP-$^{fl}$DTA$^{fl}$ mice were moved to 30 % sucrose solution after one-night of post-fixation for dehydration. The brains were embedded in OCT (Sakura, 4583) and the cryosections in 16 μm was prepared. Slices were treated with 0.5% Triton in PBS for 30 min, followed by 2 h of blocking with 2.5% BSA. Slices were incubated with primary antibodies (Supplementary Table 1) at 4 °C overnight, followed by secondary antibody (Supplementary Table 2) incubation at R.T. for 2 hs at the next day. DAPI (25 ng/ml) was used for general stain of nuclei (A10010010, Biochimica). For lipid staining, slices are secondary antibody washing, were incubated with BODIPY™ 493/503 (Invitrogen, D3922, 1 mg/ml in DMSO) diluted 1:5000 with PBS for 1 h. After three washes, the slices were mounted for further analysis.

## Image acquisition and analysis

For the overview imaging (Fig. 3B, F; Supplementary Fig. 2, Supplementary Fig. 4B, G, H), whole brain slices were scanned with fully automated slide scanner AxioScan.Z1 (Zeiss, Jena) at the AxioScan core facility in CIPMM and cell density was analyzed manually using ZEN 3.2 blue edition (Zeiss, Jena).

**Live imaging of lysosomes in vitro and ex vivo.** Cells were placed in the imaging chamber of LSM 780 confocal microscope with consistent temperature (37 °C) and CO$_2$ supply (5%). Live imaging was performed with ×63 objective (NA.1.4) for 2–5 μm z-stack with 1 μm interval. Two to four channels were imaged simultaneously with 0.8 Hz, averaging

with 2–4 acquisition, for 10–120 min. Images were analyzed with Imaris software. To track lysosome trafficking, lysosomes were labeled with Lysotracker (1:2000, L7526, Invitrogen) introduced to the medium immediately prior to the imaging. To identify lysosomal exocytosis at the contact site, Alexa Fluor® 647 Anti-LAMP1 (ab237307, Abcam) were added to the culture medium at a concentration of 1:1000 immediately prior to live imaging. The number of Alexa647$^+$ puncta within the neurons were quantified with Imaris. Neuronal somata was identified and segmented based on h3MD(Gq)-mCherry expression. Within the segment, the number of Alexa647$^+$ puncta was analyzed. In addition, these puncta were classified into two subgroups based on their distance to the contact site using Imaris software: distance <1 μm: proximal to OPC contact, distance>1 μm: distal to OPC contact.

For lysosome exocytosis ex vivo analysis, control and OPC-Cav1.2/Cav1.3 dKO mice were deeply anaesthetized with 5% isoflurane and decapitalized. Acute brain slices in 300 μm were prepared, incubated with Alexa647-anti-Lamp1 antibodies for 30 min and imaged with LSM 880 confocal microscope. Neuronal somata is segmented with Imaris and the number of Alexa647$^+$ puncta was normalized to the volume of neuronal somata. The value of the dKO group from each brain slice was normalized to the control brain slice with the same experimental settings.

**OPC morphology analysis.** Images were acquired with LSM 880 confocal microscope (Zeiss, Oberkochen) with 63x objectives (N.A. 1.4, Oil) with a 1 μm interval. OPC morphology was analyzed with 'filament function' of Imaris (Version 9.6, Oxford Instruments) with the following settings: 'autopath' selection with a maximum diameter of 10 μm and seed points with 0.3 μm; elimination of seed points within a 20 μm sphere region around starting points; and removal of disconnected segments using a 0.6 μm smooth method. Sholl analysis was performed and the distance between each circle was set at 1 μm. Subsequent to filament reconstruction, individual datasets for Sholl analysis were exported for further examination.

**Analysis of process-somata contact.** Images were acquired with LSM 880 confocal microscope (Zeiss, Jena) with 63x objectives with 1 μm interval. Three D-reconstruction of neuronal somata was conducted with Imaris. Subsequently, OPC processes were reconstructed in 'surface' mode, touching objects were split with 0.5 μm seed point diameter using morphological split, and the distance to neuronal soma was set as <0.25 μm. Following OPC process reconstruction, individual volume of contacting segment of each processes were exported for further analysis.

**Neuronal lysosome analysis.** Neuronal somata were segmented using the 'Surface' function in Imaris. Lysosomes within the somata were imaged using a STED microscope, and their absolute volume and number were analyzed with the following settings in the 'Surface' function: object detection was performed using a background subtraction with a threshold of 0.15, followed by automatic thresholding with a seed point diameter of 0.1 μm based on intensity. Seed points were filtered as 'above automatic threshold,' and no surface filtering was applied. After spot reconstruction, separate datasets for volume and the shortest distance to the OPC and neuronal surfaces were exported into individual Excel files for further analysis.

**Stimulated emission depletion (STED) microscopy.** Imaging was performed on an inverted STED Microscope (Expert Line, Abberior Instruments, Göttingen, Germany) using 488 nm, 561 nm, and 640 nm pulsed excitation lasers with respective detections at 498–520 nm, 605–625 nm, and 650–720 nm. Brain slices were visualized with a 100x silicone oil immersion objective (NA 1.4, UPLSAPO100XS, Olympus, Hamburg, Germany). STED images of lysosomes were acquired with 640 nm excitation and 775 nm STED laser with a toroidal ("donut")

depletion pattern of the STED focus. The images were recorded using Imspector software (Abberior) with a voxel size of $20 \times 20 \times 300$ nm$^3$. The pinhole size was set to 1.08 Airy Units. For 3D rendering with Imaris, image stacks were planewise linearly deconvolved (Wiener filtered) using theoretical point-spread functions and customized MATLAB codes, and further denoised with Gaussian and Median filters.

**Ultrastructural images of OPC-neuron contact and analysis.** To confirm the presence of OPC process-neuron somata contacts, we used the dataset available online at https://microns-explorer.org/ (for mouse cortex)[14] and H01 dataset at http://h01-release.storage.googleapis.com/landing.html (for human non-pathological temporal lobe)[15]. Both datasets have segmented each cell, most of which have been given identity number and cell type, thereby can be viewed using Neuroglancer. For the human data, we selected 'microglia/OPC' identified by the authors, further defining OPC identification based on their morphology. We analyzed cortical layers 2-6 individually by randomly selecting approximately 150 neurons per layer without bias. We examined PSCs in 3D using Neuroglancer. For the mouse data, we focused on 18 OPCs identified by the authors, selecting approximately 30 neurons located near these OPCs based on neuronal morphology, such as spine and dendrite characteristics. We then inspected the PSCs between OPC processes and neuronal somata using the 3D view in Neuroglancer.

## Next-generation RNA sequencing

For the RNA-seq analysis, 2 mice of the same group (ctl or dKO) were pooled and in total 6 mice were analyzed per group. The library underwent preparation and sequencing procedures conducted by Novogene, employing a series of meticulous methods. The initial quality assessment involved 1% agarose gel electrophoresis to evaluate RNA degradation and potential contamination. Subsequently, sample purity and preliminary quantitation were determined using the Bioanalyser 2100 from Agilent Technologies, USA. This instrument was also instrumental in assessing RNA integrity and final quantitation.

For library preparation, oligo d(T)25 magnetic beads were utilized to selectively isolate mRNA from the total RNA sample, employing a method known as polyA-tailed mRNA enrichment. Following this, mRNA underwent random fragmentation, and cDNA synthesis ensued using random hexamers and the reverse transcriptase enzyme. Upon completion of the first chain synthesis, the second chain was synthesized with the addition of an Illumina buffer. Through the presence of dNTPs, RNase H, and polymerase I from E. coli, the second chain was obtained via nick translation. The resultant products underwent purification, end-repair, A-tailing, and adapter ligation. Fragments of the appropriate size were enriched through PCR, introducing indexed P5 and P7 primers, with final products subjected to purification.

Verification of the library was carried out using Qubit 2.0 and real-time PCR for quantification, while the Agilent 2100 bioanalyzer was employed for size distribution detection. Quantified libraries were pooled and subsequently sequenced on the Illumina Novaseq 6000 platform, based on effective library concentration and data volume.

The qualified libraries underwent Next Generation Sequencing utilizing Illumina's Sequencing Technology by Synthesis, wherein fluorescence detection was employed for nucleotide identification during the synthesis of the complementary chain. The Novaseq 6000 sequencing system was instrumental in conducting the parallelized and massive sequencing of the libraries. The sequencing strategy employed was paired end 150 bp.

## RNA-seq data processing

The quality of RNA sequencing reads was assessed through FastQC (https://www.bioinformatics.babraham.ac.uk/projects/fastqc/). Alignment to the GRCm38 Mus musculus genome was carried out using HISAT2 v2.0.5[38] with default parameters. The gene count matrix for

each sample was generated using featureCounts v1.5.0-p3[39]. The subsequent analysis utilized the DEseq2 v1.20.0 package in R program[40]. Genes with a normalized count below 10 were excluded from downstream analysis. Significantly deregulated genes were identified with a false discovery rate below 0.05.

Differential expression analysis was performed with the 'DESeq' function with default parameters, and log fold change shrinkage was applied to the analysis results. Heatmaps depicting differentially expressed genes (DEGs) (neurons with a $p$-value < 0.05; OPCs with a $p$-value < 0.02) were visualized using pheatmap v1.0.12[41]. Selected gene set enrichment analysis for Gene Ontology (GO) and Kyoto Encyclopedia of Genes and Genomes (KEGG) pathways was conducted with ClusterProfiler v3.8.1[42]. GO enrichment analysis was conducted on significant positive genes using the hypergeometric test. Statistical significance was adjusted for multiple testing using the False Discovery Rate method (Benjamini–Hochberg procedure). GO terms from the Biological Process (BP), Cellular Component (CC), and Molecular Function (MF) with an adjusted $p$-value < 0.05 were considered significantly enriched. Significance in KEGG pathway enrichment analysis was determined by adjusted $p$-values < 0.05.

## Statistical analysis

Data were analyzed with Graphpad Prism 10.1.2 and Originpro 2022. Figures were generated with Adobe Indesign 2023/2024 and Adobe illustrator 2023/2024. For all immunostainings, two randomly selected brain slices of each mouse were used. At least three animals of each sex were analyzed per group and the data from both sexes were pooled as no sex-difference was observed. Normal distribution was tested within Graphpad Prism, and normally distributed dataset were analyzed with unpaired $t$ tests, paired $t$ test, one-way ANOVA and two-way ANOVA (indicated in each figure legend), while the Kruskal–Wallis test was used for non-normally distributed datasets. The used statistical analysis and $P$-values are indicated in the figures and legends. Data were shown as mean ± SEM. Three-five independent experiments were performed for the in vitro experiment (each experiment required one P0-1 pup for neuronal preparation and 4 P3-6 pups for OPC preparation).

## Reporting summary

Further information on research design is available in the Nature Portfolio Reporting Summary linked to this article.

## Data availability

All the data generated or analyzed in this study are included in the figures, texts, and supplementary information files. The bulk RNA seq data of OPCs and neurons generated in this study have been deposited in the GEO repository database under accession code GSE285995. Source data are provided with this paper.

## Code availability

The original R code used in this study is available on the GitHub[43] [https://github.com/Lipao-Fang/-Oligodendrocyte-precursor-cells-facilitate-neuronal-lysosome-release.git].

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

## Acknowledgements

The authors thank Hanna Pfeiffer-Unckrich and Harsha Seerapu for their excellent technical support and Daniel Schauenburg for excellent animal husbandry for experimental assistance. We are also grateful to Dr. Manuel Buttini (University of Luxembourg, Luxembourg) for providing brain slices of Tg2576 and corresponding control mice; Prof. Jaqueline Trotter and Prof. Eva-Maria Krämer-Albers (University of Mainz, Mainz, Germany) for providing the NG2 antibody and sharing OPC culture protocol, respectively; JoAnn Buchanan and Nuno Maçarico da Costa (Allen Institute for Brain Sciences, Seattle, WA 98109) for the introduction to the electron microscopy database of mouse cortex. This work was supported by grants from the Chica and Heinz Schaller Research Foundation and the Deutsche Forschungsgemeinschaft: SFB1158 (A09) to A.A.; the European Commission (H2020-MSCA-ITN EU-GliaPhD to F.K.); the BMBF (EraNet-Neuron BrIE to F.K.), Deutsche Forschungsgemeinschaft (SPP 1757and SFB 1158 (A09) to F.K.; BA 8014/1-1 to X.B.), University of Saarland (GradUS global2019 and 2020 to L.-P.F.; HOMFORexzellent2020 and NanoBioMed Young Investigator grant 2020 to H.-F.C.; Mini-proposal of SFB1027 to R.Z.; HOMFOR-exzellent2018 and NanoBioMed Young Investigator grant 2021 to X.B.).

## Author contributions

L.-P.F and X.B. conceived the project with input from F.K.; L.-P.F., Z.W., Y.S., N. Z., F.F., W.H., A.A., C.Y., J.N., and X.B. treated animals, prepared brain slices, performed immunostainings and L.-P.F., Z.W., J.N., A.S., and X.B. performed imaging and analyzed data; L.-P.F., S.B., and X.B. performed EM data analysis; C.-H. L., S.-M. T., and H.-F. C. performed cell culture and L.-P.F performed imaging and data analysis; Y.M., M.A.L., L.-P.F. and performed STED imaging and ex vivo imaging; N.Z., W.H. and S.B performed preliminary experiments; R.Z. provided technical support and guide for image analysis with Imaris; D.G. performed calcium signal analysis; Q.G. and X.B. performed intracortical virus injection; A.W., V.W., and U. B. produced AAV viruses; F.K. and X.B. supervised the project, ensured the coordination and resource support of the project; X.B. wrote the manuscript with comments of the other authors.

## Funding

## Competing interests
