## [Transparent Peer Review file · Nature Communications]

Oligodendrocyte precursor cells facilitate neuronal lysosome release

Corresponding Author: Dr Xianshu Bai

This manuscript has been previously reviewed at another journal that is not operating a transparent peer review scheme. The manuscript was considered suitable for publication without further review at Nature Communications.

Version 0:

Reviewer comments:

Reviewer #2

(Remarks to the Author)

The authors have significantly revised the manuscript and eliminated their earlier claim that lysosomes are transferred from neurons to OPCs. They now focus on lysosome exocytosis that seems to be promoted by neuronal activity and OPC contact. Overall, the manuscript is much more focused, and the conclusions are more adequately supported by the data. The Cav double knockout approach is still rather weak, but the DTA-mediated OPC ablation experiment provides additional support. The major concerns have been adequately addressed. Please make sure multiple biological replicates and not just technical replicates, are taken for all the experiments.

Reviewer #3

(Remarks to the Author)

The authors have removed the data claiming lysosomal transfer from one cell to another. The revised version now appears solid and interesting. I support the publication of this version of the paper.

Point-to-point responses to reviewers' comments:

Reviewer #2 (Remarks to the Author):

The authors have significantly revised the manuscript and eliminated their earlier claim that lysosomes are transferred from neurons to OPCs. They now focus on lysosome exocytosis that seems to be promoted by neuronal activity and OPC contact. Overall, the manuscript is much more focused, and the conclusions are more adequately supported by the data. The Cav double knockout approach is still rather weak, but the DTA-mediated OPC ablation experiment provides additional support. The major concerns have been adequately addressed. Please make sure multiple biological replicates and not just technical replicates, are taken for all the experiments.

Thank you for the positive comments. The biological replicates, particularly for the in vitro experiments, are not just technical replicates, but also biological replicates. We have indicated this information in the figure legends as well as in the Methods section.

For primary cultures, we required 1 pup per neuronal culture and 4 pups per OPC culture. We repeated at least three times the experiments, and every replicate is from different biological individual.

Reviewer #3 (Remarks to the Author):

The authors have removed the data claiming lysosomal transfer from one cell to another. The revised version now appears solid and interesting. I support the publication of this version of the paper.

We thank the reviewer for the positive comments.